# A multicellular way of life for a multipartite virus

Anne Sicard[1], Elodie Pirolles[1], Romain Gallet[1], Marie-Stéphanie Vernerey[1], Michel Yvon[1], Cica Urbino[1,2], Michel Peterschmitt[1,2], Serafin Gutierrez[1], Yannis Michalakis[3], Stéphane Blanc[1]*

[1]BGPI, INRA, CIRAD, Montpellier SupAgro, Université de Montpellier, Montpellier, France; [2]CIRAD, BGPI, INRA, Montpellier SupAgro, Université de Montpellier, Montpellier, France; [3]MIVEGEC, CNRS, IRD, Université de Montpellier, Montpellier, France

**Abstract** A founding paradigm in virology is that the spatial unit of the viral replication cycle is an individual cell. Multipartite viruses have a segmented genome where each segment is encapsidated separately. In this situation the viral genome is not recapitulated in a single virus particle but in the viral population. How multipartite viruses manage to efficiently infect individual cells with all segments, thus with the whole genome information, is a long-standing but perhaps deceptive mystery. By localizing and quantifying the genome segments of a nanovirus in host plant tissues we show that they rarely co-occur within individual cells. We further demonstrate that distinct segments accumulate independently in different cells and that the viral system is functional through complementation across cells. Our observation deviates from the classical conceptual framework in virology and opens an alternative possibility (at least for nanoviruses) where the infection can operate at a level above the individual cell level, defining a viral multicellular way of life.
DOI: https://doi.org/10.7554/eLife.43599.001

*For correspondence: stephane.blanc@inra.fr

**Competing interests:** The authors declare that no competing interests exist.

## Introduction

Viruses introduce their genome into a susceptible cell and highjack diverse cell functions to complete their replication cycle. Once this 'cell-autonomous' cycle is completed, virus particles exit the infected cell and enter a new healthy one where a similar cycle is reiterated. This universal view of the viral way of life has been applied to multipartite viruses since their discovery, over half a century ago (*Brakke et al., 1951*; *Lister, 1966*; *van Kammen and van Griensven, 1970*; *Gokhale and Bald, 1987*), despite the fact that such a conceptual framework fails to explain the evolution and even the functioning of these viral systems (*Iranzo and Manrubia, 2012*; *Sicard et al., 2016*; *Lucía-Sanz and Manrubia, 2017*).

Depending on the viral species, multipartite viruses have their genome composed of two to eight segments of DNA or RNA (single or double stranded), each encapsidated individually in a separate virus particle. They have been reported to infect frequently plants and fungi where they represent at least 35–40% of the viral genera and families described (*Hull, 2014*), rarely insects (*Hu et al., 2016*; *Ladner et al., 2016*), and only hypothetically vertebrates (*Ladner et al., 2016*). The benefits of the individual encapsidation of distinct segments are unclear and debated (*Sicard et al., 2016*; *Lucía-Sanz and Manrubia, 2017*). In contrast, the reduced probability to infect individual host cells with at least one copy of each segment is unanimously acknowledged as a major cost that increases with the number of segments composing the viral genome (*Iranzo and Manrubia, 2012*). This cost appears insurmountable, at least for viruses with more than three genome segments, and sets the existence of such highly multipartite viruses as an enigma in general biology (*Iranzo and Manrubia,*

**eLife digest** Many viruses are small particles consisting of genetic material surrounded by a coat made of proteins. They are unable to multiply on their own and so they must enter a host cell and trick it into reading their genetic information to produce new virus particles.

It is generally thought that the process of making new virus particles happens independently in each infected cell. This idea assumes that a given particle contains the entire set of genetic material (known as the genome) of that virus, but this is not always the case. Many so-called 'multipartite' viruses have genomes that are split into several segments carried in separate particles: in this case, a single particle only contains a portion of the entire viral genome.

Faba bean necrotic stunt virus (or FBNSV for short) is a multipartite virus that infects and causes disease in members of the pea and bean family. There are eight types of FBNSV particle that each carries a distinct genome segment, a small section of the entire viral genome. There is a low probability that a single cell could be infected with all eight different types of particle at the same time and receive the complete FBNSV genome. So how is this virus able to successfully multiply within a plant?

To address this question, Sicard et al. used microscopy to study FBNSV genome segments as they infected the cells of faba bean plants. The experiments confirmed that the eight different segments of the FBNSV genome were not necessarily found together within the same cell, but instead accumulated independently in different cells. This means that a cell infected with FBNSV may be unable to make all of the proteins needed to assemble new virus particles. However, additional experiments demonstrated that infected cells may be exchanging virus proteins, which could enable them to create complete virus particles.

The findings of Sicard et al. demonstrate that FBNSV hijacks groups of host cells to manufacture new virus particles, rather than relying on individual cells as previously thought. It is possible that other multipartite and non-multipartite viruses work a similar manner. Ultimately, this knowledge may reshape what we know about how viruses infect their hosts.

DOI: https://doi.org/10.7554/eLife.43599.002

*2012*; *Sicard et al., 2016*; *Lucía-Sanz and Manrubia, 2017*). Nevertheless, its empirical basis has never been questioned.

Using a highly multipartite virus with eight genome segments, we here propose that the conceptual framework in Virology should be amended to account for such viral systems. Indeed, in our specific experimental model species, we demonstrate that the distinct segments do not need be together in individual cells for the system to be functional. They accumulate independently in and complement across neighboring cells, defining a multicellular way of life.

## Results

### Distinct viral genome segments do not necessarily co-localize in individual infected cells

Our first aim was to experimentally verify whether the different segments of a multipartite virus have to be together in individual cells for the system to be functional. We used the highly multipartite faba bean necrotic stunt virus (FBNSV, genus *Nanovirus*, Family *Nanoviridae*), whose genome is composed of eight circular ssDNA segments (*Figure 1A*) (*Grigoras et al., 2009*). We first compared the localization of distinct segments in individual cells of faba bean host plants with specific fluorescent probes and confocal microscopy. In plants submitted to productive viral infection (see Materials and methods), the visualization of pairs of segments respectively labeled with green and red fluorochromes immediately revealed that distinct segments do not necessarily co-occur but are most often found in different cells (*Figure 1B–I* and *Figure 1—figure supplement 1*). We tested 7 out of the 28 possible pairs of segments. In all cases we observed a similar situation with one segment most often highly accumulated in the absence of the other in individual cells. Remarkably, this applied even to the pairs from the three segments encoding for the three basic functions of plant viruses: replication (segment R encoding M-Rep), encapsidation (segment S encoding CP) and intra-

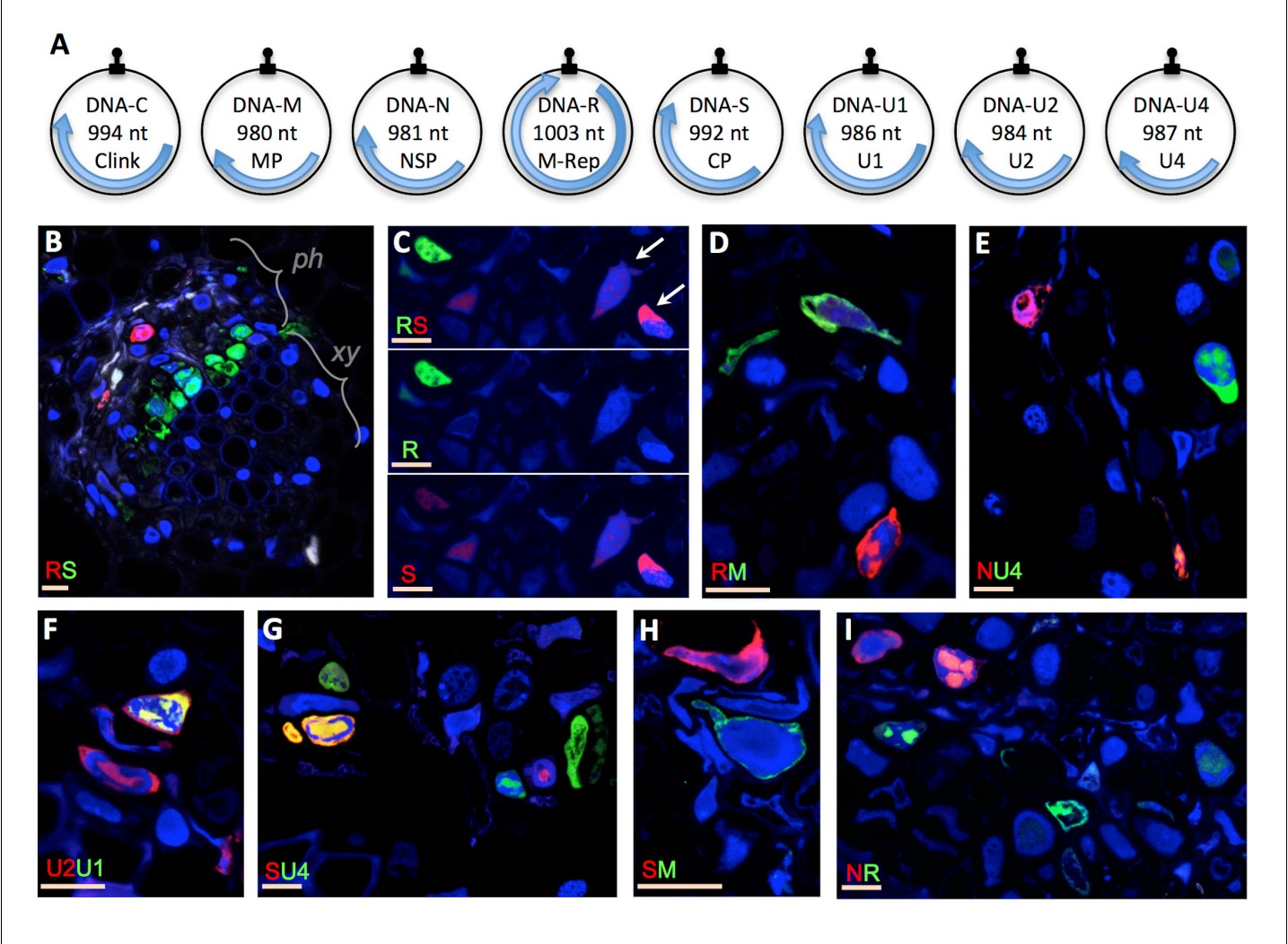

**Figure 1.** Localization of distinct FBNSV segments in individual cells. The FBNSV genome comprises eight single-stranded circular DNA segments (**A**). The size and name of the segments and encoded proteins are indicated inside the circles. Black stem-loops and blue arrows indicate the replication origins and the coding regions, respectively. The function of each segment is described in *Grigoras et al. (2009)*: DNA-C, cell cycle resetting; DNA-M, within plant viral movement; DNA-N, nuclear shuttle protein; DNA-R, replication; DNA-S, coat protein/encapsidation; DNA-U1, U2 and U4, unknown. B-I show cross sections of infected faba bean petioles, with phloem (*ph*, the only tissue infected by nanoviruses), and xylem (*xy*) bundles indicated in B. Pairs of segments are green- and red-FISH probed as indicated in each micrograph. The fluorochromes incorporated into the probes are inverted in B and C for control. Color channels are merged in all images but C where green and red are shown merged and separated to evidence the accumulation of the segment S in cells where the segment R is not detected (exemplified by white arrows). Split channel images corresponding to images B and D-I are shown in *Figure 1—figure supplement 1*. Nuclei are DAPI-blue stained. Horizontal bars = 10 microns.
DOI: https://doi.org/10.7554/eLife.43599.003

The following figure supplements are available for figure 1:

**Figure supplement 1.** Split color channel images corresponding to images of *Figure 1*.
DOI: https://doi.org/10.7554/eLife.43599.004

**Figure supplement 2.** Co-occurrence of segments in individual cells does not depend on time of infection.
DOI: https://doi.org/10.7554/eLife.43599.005

host movement (segment M encoding MP), suggesting that cells containing all eight segments are extremely rare.

Though challenging the current view of the viral cell-autonomous replication cycle, a simple explanation of our observation is that the FBNSV can function while its genome segments occur in distinct neighboring cells. This possibility calls i) for further evidence that the accumulation of a given segment is independent of the accumulation of the others in individual cells and ii) for the proof of

concept that the function encoded by a given viral segment can complement the others at a distance, in cells where this segment is absent.

## Distinct segments accumulate independently in distinct individual host cells

One may argue, sticking to the paradigm that the viral genetic information is replicated as an integral genome within individual cells, that all FBNSV segments are present in infected cells but that the apparent absence of some in *Figure 1* is due to the detection limit of our technique. Because every technique has its limit, whatever the technology implemented, it would not be reasonable to certify that the absence of detection is a proof of the absence of the corresponding segment. We thus imagined an approach where the detection limit becomes irrelevant. For a given pair of segments, we quantified and compared both green and red fluorescence in all individual cells where at least one of the two was observed above background (for detailed quantification procedure see the Materials and methods section). By doing so, we alleviate the problem of the limit of detection and rather question whether the accumulation of one segment of the pair is dependent on that of the other. As a positive control of this approach, we first produced two fluorescent probes, each specifically labeling a different region of the same segment R (probes R1 and R2). *Figure 2* illustrates that all cells labeled by R1 are also labeled by R2 (*Figure 2A*). Moreover, plotting the average intensity of green over red fluorescence for each of these cells resulted in a highly significant linear relationship (correlation coefficient r = 0.90, p=1.98 $10^{-23}$ in the example of petiole N° 42 in *Figure 2B*). Four independent repeats of this control similarly showed strong correlations (*Table 1*). This result validates our approach by demonstrating that when we monitor two viral DNA sequences whose

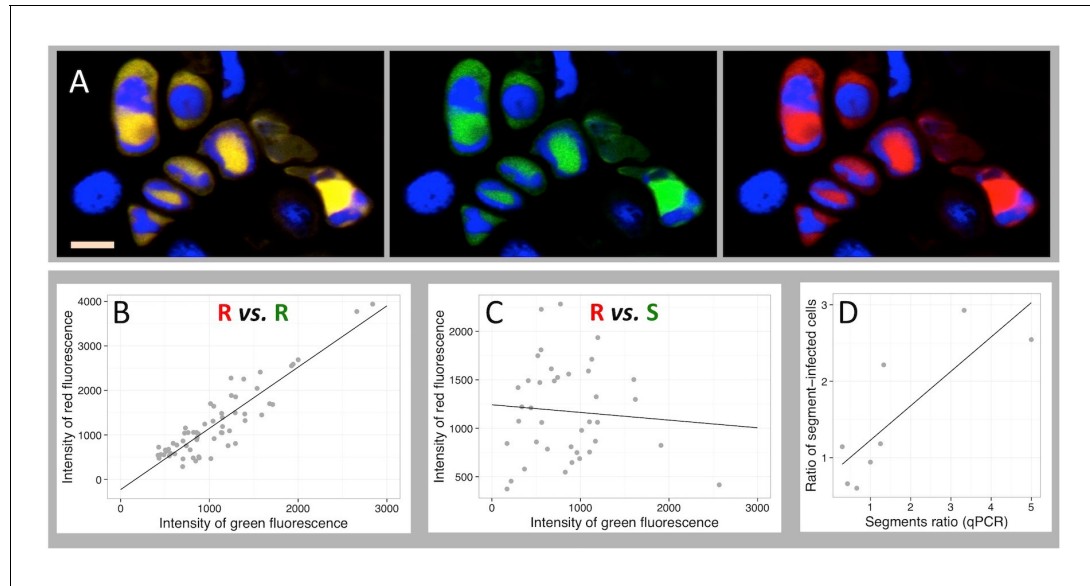

**Figure 2.** Independent accumulation of FBNSV segments in individual cells. (**A**) labeling of two distinct regions of the same segment R by a red and green fluorescent probe, respectively. Green and red channels are shown merged (left) and separated (center and right). Nuclei are DAPI-stained in blue. Horizontal bar is 10 microns. The average intensity of the green fluorescence within individual cells is plotted against that of the red in B and C. Green and red probes are targeted against two distinct regions of the same segment R in (**B**), and against two distinct segments, respectively S and R in (**C**). The analyzed cells are from petiole N°42 for the positive control in (**B**) and from petiole N°5 for the pair R/S in (**C**). The same analysis from three additional petioles for the control (also see *Figure 2—figure supplement 1*) and nine additional petioles for the segments pair R/S are summarized in *Table 1*. Panel D plots the ratio of the frequency of two segments of a pair (estimated by qPCR on total DNA extracted from infected tissues) against the ratio of the number of cells they respectively infect within eight petioles from eight different infected plants (data provided in *Supplementary file 1*: Table S1).

DOI: https://doi.org/10.7554/eLife.43599.006

The following figure supplement is available for figure 2:

**Figure supplement 1.** Reliability of the fluorescence quantification method.

DOI: https://doi.org/10.7554/eLife.43599.007

**Table 1.** Correlation test for the accumulation of two segments in individual cells

| Petiole[*] | 1 | 2 | 3 | 4 | 5 | 7 | 9 | 10 | 11 | 12 | 13 | 14 | 15 | 16 | 18 | 40 | 41 | 42 | 43 |
|---|---|---|---|---|---|---|---|---|---|---|---|---|---|---|---|---|---|---|---|
| dpi[*] | 25 | 12 | 12 | 12 | 27 | 11 | 25 | 25 | 25 | 20 | 20 | 20 | 20 | 20 | 20 | 19 | 21 | 19 | 27 |
| Pair[†] | R/S | | | | | | | | | | S/M | | | R/M | | R1/R2 | | | |
| n[‡] | 47 | 18 | 67 | 31 | 39 | 28 | 20 | 29 | 39 | 15 | 19 | 43 | 32 | 23 | 46 | 16 | 22 | 62 | 49 |
| F[§] | 3.04 | 0.19 | 0.15 | 1.23 | 0.24 | 0.46 | 0.10 | 0.69 | 0.71 | 1.94 | 4.73 | 1.74 | 2.25 | 2.68 | 2.43 | 1323 | 297 | 254 | 345 |
| r[¶] | 0.25 | 0.11 | 0.05 | −0.20 | −0.08 | −0.13 | 0.08 | −0.16 | 0.14 | 0.36 | −0.47 | 0.20 | 0.26 | −0.34 | 0.23 | 0.99 | 0.97 | 0.90 | 0.94 |
| p[††] | 0.09 | 0.66 | 0.70 | 0.28 | 0.63 | 0.51 | 0.75 | 0.41 | 0.40 | 0.19 | 0.04 | 0.19 | 0.14 | 0.12 | 0.13 | $2.89^{e-15}$ | $1.81^{e-13}$ | $1.98^{e-23}$ | $2.81^{e-23}$ |

[*] Code number of petioles and time of sampling (dpi: days post infection).

[†] Pair of segments analyzed for each petiole.

[‡] n = number of cells detected positive for at least one of the segments of the pair. Fluorescence quantification for each of these cells is given in **Supplementary file 2**: Table S4.

[§] Value of F from regression analysis.

[¶] Correlation coefficient.

[††] p-value.

DOI: https://doi.org/10.7554/eLife.43599.008

accumulation should be highly correlated, such as two regions of the same segment, we indeed find that they co-localize and accumulate at highly correlated levels.

This approach was then extensively used for the pair of segments R/S because they encode the two key viral functions, replication and encapsidation. In total, we tested 10 petioles, each from a different plant infected independently. No significant correlation could be found (*Figure 2C*, *Table 1*), whether at early or late stages of infection (*Table 1* and also see *Figure 1—figure supplement 2*), showing that the accumulation of the segment S in a cell is independent of that of the replication-encoding segment R. We extended this analysis to five additional petioles from five independent plants, two for the pair R/M and three for the pair S/M, and again found no significant positive correlation; a result indicative of independent accumulation (*Table 1*). To further support this conclusion, we used the earlier reported variation in the relative frequency of the segments across infected plants (*Sicard et al., 2013*). Within an infected petiole, if two segments independently enter and accumulate in individual cells, then their relative frequencies in infected tissues should be proportional to the ratio of the number of cells they respectively infect. We quantified by qPCR the relative frequency of the two segments of the pair in a fragment of eight of the analyzed petioles (*Supplementary file 1*: Table S1). *Figure 2D* shows a significant correlation between the relative frequency of the segments in the tissues and the relative number of cells they respectively infect (n = 7, F = 12, R = 0.677, p-value=0.0122). Hence, two different technologies (FISH-confocal microscopy and quantitative real-time PCR) concur to demonstrate that the accumulation of distinct FBNSV segments in individual cells is independent.

## The function of a viral gene can act in a cell where the gene itself is absent

To make sense out of the above observation we assumed that a viral function can be present and act in a cell where its encoding segment is not, and thus that the viral system can be functional despite the independent dispersal of its distinct genes in distinct cells. The validation of this assumption was set up with a focus on the replication function. We first confirmed that the segments detected in individual cells are the product of replication. Then we searched for the presence of the protein M-Rep in these cells, to support replication even when its encoding segment R is absent.

Two possible alternatives could lead to the detection of a segment as a fluorescent signal spread out in a large volume of the nucleus. Either one (or very few) copy entered, replicated and invaded the nucleus to detectable level, or a number of copies large enough to be detectable entered and diffused throughout the nucleus with no replication. When looking at images shown in *Figure 1*, the latter appears unrealistic because it would imply the specific sorting of different segments at their entry in distinct nearby cells. Nevertheless, to experimentally dismiss this possibility, we similarly investigated the distribution of two alleles of the same segment (differing solely by a small inserted

marker sequence) for which no specific sorting can be expected. In the absence of sorting, if the segment massively enters cells to detectable level without replication, the two alleles should be co-detected in most cells. In contrast, if one (to very few) copy of this segment enters each individual cell and then accumulates through replication, allele-specific FISH labeling should reveal solely one of the two alleles in most cells. Three independent experiments were carried out for each of the segments S and N where two genetic markers have earlier been inserted (*Gallet et al., 2017*). They consistently revealed that most individual cells (54% to 100%) contained only one detectable allele (*Figure 3—figure supplement 1*), confirming that one to very few copies of the segment initially enter individual cells and thus that detection all over the nucleus can only result from replication.

We then confronted the cells where the segment S is detectable (thus where it has replicated) to the detection of either segment R or its protein product M-Rep, by a combination of FISH and immunofluorescence-labeling (*Figure 3A and B*). While segment R was detectable in only a minority of these cells (approx. 40%), its protein product M-Rep was positively revealed in nearly 85% (*Figure 3C*), indicating that the expression product of R segment can move to neighboring cells. That the protein M-Rep is undetectable in a few cells replicating segment S (approx. 15%) can be explained by distinct turnovers for the genome segments and their expression products. Indeed, both mRNAs and proteins have a relatively rapid turnover within a cell, whereas the DNA viral sequences can be stored indefinitely into the cell nucleus either encapsidated or as stable minichromosomes (*Gronenborn, 2004*; *Ramesh et al., 2017*; *Deuschle et al., 2016*; *Rodríguez-Negrete et al., 2014*). Thus, we assume that for all cells where at least one FBNSV segment is detectable the protein M-Rep is or has been present, but that this protein may have disappeared in

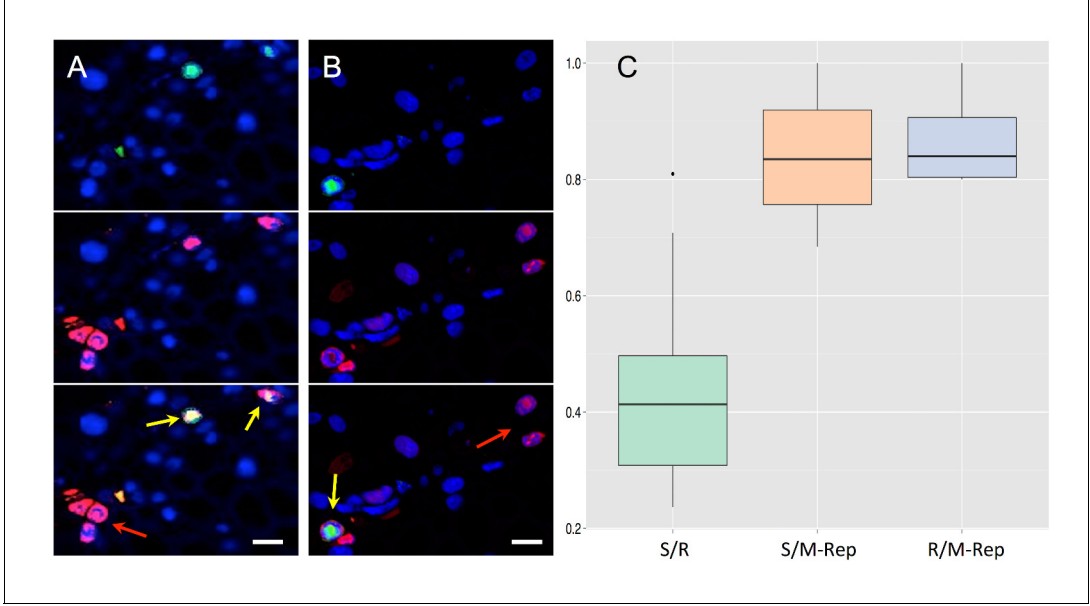

**Figure 3.** Accumulation of the protein M-Rep in cells containing either segment S or segment R. Segment R (**A**) or S (**B**) is FISH-labeled with a green fluorescent probe whereas the M-Rep protein is immuno-labeled with a red fluorochrome. Green and red channels are shown separated (top and middle) and merged (bottom). Nuclei are DAPI-stained in blue. Horizontal bar is 10 microns. Yellow arrows point at cells where both the DNA segment and the protein M-Rep are detected, whereas red arrows exemplify cases where only M-Rep is detectable. The boxplots (**C**) are constructed from *Supplementary file 1*: Table S1. The proportion of cells with the S segment that also contain the R segment (green left boxplot, estimated from 10 petioles) is compared to the proportion of cells with the S segment that also contain the protein M-Rep (middle orange boxplot, estimated from 14 petioles). The proportion of individual cells with detectable R segment that also contain the protein M-Rep was estimated from 4 infected petioles (blue right boxplot). Difference between left green boxplot and orange middle boxplot is highly significant (GLM model, p-value ($>\chi_1^2$) = 2.94$^{e-12}$); that between middle orange boxplot and right blue boxplot is not (GLM model, p-value ($>\chi_1^2$) = 0.55).

DOI: https://doi.org/10.7554/eLife.43599.009

The following figure supplement is available for figure 3:

**Figure supplement 1.** Distinct copies of the same segment accumulate in distinct cells.

DOI: https://doi.org/10.7554/eLife.43599.010

some cases at the moment of the observation. Consistent with this assumption is the observation of a similar proportion of cells containing the segment R where its own expression product M-Rep is no longer detectable (*Figure 3C*).

It is important to note the detection of the M-Rep protein, often with intense signal (*Figure 3A & B*), in a large number of cells where segment R is absent (*Supplementary file 1*: Table S1, petioles 36–39). Moreover, *Figure 3C* (orange middle and bleu-grey right boxplots) indicates that the protein M-Rep is not more associated to cells containing its own encoding segment R than to cells containing other segments. Although they represent indirect evidence, these observations together further support our conclusion that either the mRNA or the protein M-Rep itself can travel from the producing cells (those where segment R accumulates) to other cells of the host, as further discussed below.

## Discussion/conclusion

Altogether, our results demonstrate that key genome segments of the FBNSV accumulate in individual cells independently of the presence and accumulation of the others. We thus propose that the different parts of the viral genome can complement across distinct neighboring cells and can only sustain the productive infection at a multicellular tissue level. As numerous other plant virus species (*Hipper et al., 2013*; *Folimonova and Tilsner, 2018*), nanoviruses are restricted to vascular tissues and replicate in phloem companion and parenchyma cells (*Shirasawa-Seo et al., 2005*). A model compatible with our observations is that each genome segment entering and expressing within a cell can export its gene products as 'common goods' in neighboring cells and/or sieve elements, so that these common goods are redistributed among interconnected cells and complement the system. The demonstration that the protein M-Rep accumulates and functions in cells devoid of its encoding DNA-R is a proof of concept fully supporting this model. Numerous studies have shown that plant mRNAs (*Kehr and Kragler, 2018*) and proteins can move from cell to cell or long distance (*Turnbull and Lopez-Cobollo, 2013*; *Lopez-Cobollo et al., 2016*; *Paultre et al., 2016*), even from root to shoot. Unfortunately, the mechanisms and specific control of these mRNAs and proteins mobility is vastly elusive. For FBNSV, each segment may export and redistribute its product in a distinct way, some may produce mobile mRNA, others may produce mobile protein, and others may be complemented for movement by the product of a specific segment (or even by a host factor) that may act as a carrier. Specifying these diverse possibilities is beyond the scope of this report. Likewise, whether this non-cell-autonomous model can be extended to viral species invading non-phloem tissues where cell communication is more restricted is unknown. Non phloem-restricted viruses could induce the formation of symplastic domains, either by manipulating the endogenous capacity of the host plant to do so (*Faulkner, 2018*) or by opening plasmodesmata through the action of their non-cell-autonomous movement protein (*Lucas, 2006*), but this possibility awaits further investigation. Here we introduce an additional concept in virology, which is compatible with empirical observations and which partially alleviates the insurmountable cost in highly multipartite viral systems such as FBNSV (*Iranzo and Manrubia, 2012*): because concomitant infection of individual cells by all genomic segments is not necessary, the associated putative cost should be much smaller if not nil at the within-host level. We earlier discussed the fact that the analogous cost upon between-host transmission and the mechanisms of its compensation remain to be understood (*Gallet et al., 2018*).

Such a multicellular way of life could be adopted in other multicomponent genetic entities, such as other multipartite viruses, segmented viruses which often fail to encapsidate all genome segments together (*Luque et al., 2009*; *Wichgers Schreur and Kortekaas, 2016*; *Brooke, 2017*), satellites, and defective interfering particles, if functional complementation could occur at a supra-cellular level.

## Materials and methods

### Key resources table

| Reagent type or resource | Designation | Source/reference | Identifiers | Additional information |
|---|---|---|---|---|

*Continued on next page*

*Continued*

| Reagent type or resource | Designation | Source/reference | Identifiers | Additional information |
|---|---|---|---|---|
| Gene (virus) | FBNSV JKI-2000 Segment C | GenBank | GenBank: GQ150780.1 | complete sequence |
| Gene (virus) | FBNSV JKI-2000 Segment M | GenBank | GenBank: GQ150781.1 | complete sequence |
| Gene (virus) | FBNSV JKI-2000 Segment N | GenBank | GenBank: GQ150782.1 | complete sequence |
| Gene (virus) | FBNSV JKI-2000 Segment R | GenBank | GenBank: GQ150778.1 | complete sequence |
| Gene (virus) | FBNSV JKI-2000 Segment S | GenBank | GenBank: GQ150779.1 | complete sequence |
| Gene (virus) | FBNSV JKI-2000 Segment U1 | GenBank | GenBank: GQ150783.1 | complete sequence |
| Gene (virus) | FBNSV JKI-2000 Segment U2 | GenBank | GenBank: GQ150784.1 | complete sequence |
| Gene (virus) | FBNSV JKI-2000 Segment U4 | GenBank | GenBank: GQ150785.1 | complete sequence |
| Strain, strain background (nanovirus, FBNSV) | FBNSV infectious clone | reference N° 11 | Isolate FBNSV-[ET:Hol:97] | |
| Antibody | Anti-M-Rep, rabbit, polyclonal | reference N°32 | FBNYV-M-Rep 8th Bleed | Dilution 1/300 |
| Sequence-based reagent | Synthetic short probes | this paper, synthetsized by the company Eurogenetec, Liège, Belgium | Nmys2-Red, Nmys7-Green, Smys1-Red, Smys8-Green | see Table S2 |
| Sequence-based reagent | random primed probe for distinct segments | this paper | | probes for distinct segments were prepared as described in the Materials and methods section |
| Sequence-based reagent | Primers to produce the templates for the probes | this paper, synthesized by the company Eurogenetec, Liège, Belgium | | see Table S2 |
| Commercial assay or kit | Bioprime DNA-Labeling system kit | Invitrogen, Carlsbad, CA, USA | reference product: 18094011 | see Materials and methods section |
| Other: Host plant | Faba bean | Vilmorin, Saint Quentin Fallavier, France | Vicia Faba, cultivar 'Seville' | |

## Host plant, virus isolate and inoculation

Faba bean (*Vicia faba*, cv. 'Sevilla') was used as the host plant in all experiments, and seeded, maintained and inoculated as described (*Sicard et al., 2013*; *Gallet et al., 2017*; *Gallet et al., 2018*; *Sicard et al., 2015*). The viral infectious clone is from the species *Faba bean necrotic stunt virus*, genus *Nanovirus*, family *Nanoviridae*. Its construction in agrobacterium plasmids has been earlier described in details (*Grigoras et al., 2009*), as well as all conditions and procedures for agro-inoculation (*Grigoras et al., 2009*; *Sicard et al., 2013*; *Gallet et al., 2017*; *Gallet et al., 2018*; *Sicard et al., 2015*). Plants were sowed as one seed per pot. The plantlets were agro-inoculated nine days later and symptoms appeared from 9 days post inoculation (dpi) till 19 dpi, depending on individual plants. As shown in all the publications cited in this paragraph, the inoculation of plants with this infectious clone routinely yields a productive viral infection, meaning that the FBNSV particles can be purified from these infected plants and very efficiently transmitted to new healthy ones by aphids.

## Sampling of infected material

Infected plants were sampled at different days post inoculation: at 11, 12, 13, 17, 20, 25 and 27 dpi (*Table 1* and *Supplementary file 1*: Table S1). Nanoviruses, and thus FBNSV, replicate and accumulate primarily in phloem vascular bundles of the upper part of the plant, in companion cells and phloem parenchyma cells of the stem, petioles and main leaf veins. A 1 to 3 cm-long section of the petiole of the upper leaf level was cut off from each infected plant and immediately fixed and processed for fluorescent in-situ hybridization (FISH) and/or immunofluorescence labeling (IF). In some cases, the petiole was divided into two equal portions, one for FISH and/or IF and the other for DNA extraction and qPCR.

## Quantitative real-time PCR

The FBNSV genome is composed of eight circular ssDNA segments of around 1 kb each (*Figure 1A*). Total DNA extraction from infected petioles and qPCR reactions specific to each of the eight segments were performed as described previously (*Gallet et al., 2017*), including primer pairs, PCR conditions, and post-PCR data analysis. The relative frequency of two segments of a pair within a given petiole was obtained by dividing the estimated copy number of a segment by that of the other.

## Fluorescent in situ hybridization and immunofluorescence labeling

Random priming and incorporation of Alexa Fluor-labeled dUTP (either Alexa Fluor 488 or Alexa Fluor 568 for green and red labeling, respectively) were performed to prepare probes specific to each of the eight segments, with the BioPrime DNA labeling system kit (Invitrogen) and according to the manufacturer instructions except that dUTP-Alexa Fluor were used in place of dCTP-biotine of the kit. For each probe, the template DNA corresponded to a PCR-amplified fragment restricted within the viral gene coding sequence, which is the sequence unique to each segment. The primer pairs used to amplify the coding sequence of the targeted segments are listed in *Supplementary file 1*: Table S2. In some cases, short probes were synthesized by the company Eurogentec and covalently linked to either ATTO-488 (green) or ATTO-565 (red) fluorochrome (sequences also available in Table S2). The specificity of each probe was tested on membranes, on dotted plasmids each containing one of the eight viral segments, as well as on healthy plants.

Each petiole harvested from an infected plant as described above was fixed by gentle stirring overnight at 4°C in PBS buffer containing 4% paraformaldehyde and 0.2% Tween-20. After one rinse with 2 mL of PBS buffer containing 0.1M glycine, the petiole was transferred into ethanol 70% and stored at 4°C until use (max. storage time: 1 month).

Fixed petioles were then placed individually into Eppendorf tubes containing 8% low-melting agarose in water at 40°C. Petiole trunks were maintained into an upright position until the agarose cooled down and polymerized, and placed at 4°C for 4 hr. The jellified agarose blocks were extorted from the tubes and cross-sections of 80–100 microns were produced with a Vibratome HM650V (Microm), set up in mode CPC, program 50, speed 23, frequency 100, and amplitude 0.6. Cross-sections were treated in 1 mL of Carnoy solution (six volumes chloroform, three volumes ethanol, and 1 vol acetic acid) during 1 hr under gentle stirring and then rinsed once in PBS buffer. An RNAse treatment (100 µg/mL of RNAse1 in PBS) was applied to all samples for 45 min. at 37°C, in order to eliminate viral mRNAs, followed by one rinse in PBS and three additional rinses in hybridization buffer (20 mM Tris-HCl pH8, 0.9M NaCl, 0.01% SDS, 30% Formamide).

Fluorescent probes were diluted thirty times in hybridization buffer (10 µL in 300 µL total), denatured 10 min. at 100°C and then rapidly cooled on ice for 15 min. Petiole cross sections were then incubated overnight at 37°C in the diluted and heat-denatured probes solutions into embryo dishes sealed with parafilm membranes. After three rinses of 5 min with hybridization buffer and one with PBS, petiole sections were mounted on microscopy slides in Vectashield antifade mounting medium containing 1.5µg/mL DAPI for staining nuclei.

Some samples were further treated for immuno-fluorescent labeling of M-Rep protein. In these cases the FISH-treated petiole cross sections were collected after the last PBS rinse and blocked for 1 hr and 30 min. in PBS + 5% BSA. The incubation with the primary antibody (M-Rep specific antibody 'FBNYV-M-Rep 8[th] bleed' (*Vega-Arreguín et al., 2005*), diluted 1/300) was in PBS + 5% BSA overnight at 4°C, and that with the secondary antibody (goat anti-rabbit Alexa Fluor 594 IgG

conjugate, diluted 1/250, Life Technologies) was in PBS + 5% BSA for 1 hr at 37°C. Samples were submitted to three rinses of 10 min each in PBS + 0.05%Tween-20 at room temperature after incubations with primary and secondary antibodies. Samples were finally transferred into PBS and then mounted on microscopy slides as above.

## Confocal microscopy observations and fluorescence quantification

Observations were all performed in sequential mode using a Zeiss LSM700 confocal microscope. Alexa Fluor 488 and ATTO 488 were excited with a 488 nm laser and the variable secondary dichroic (VSD) beam splitter was set to recover fluorescence up to 535 nm. Alexa Fluor 568/594 and ATTO 565 were excited with a 555 nm laser and the VSD was set to recover fluorescence up to 615 nm. DAPI was excited with a 405 nm laser and the VSD was set to recover fluorescence up to 626 nm, with the additional use of a short pass SP490 eliminating wave lengths > 490 nm. Images were acquired with 40x or 63x objectives at variable resolutions, depending on their intended use, with a pinhole aperture of 1 airy unit. Acquisitions were either in plane or stack mode, and the images used from stacks correspond to maximum intensity projections. The settings for the acquisition of images shown in *Figures 1*, *2* and *3*, *Figure 1—figure supplement 1* and *Figure 3—figure supplement 1* are detailed in *Supplementary file 1*: Tables S3. All images used for fluorescence quantification were acquired with the 40x objective, and the resolution chosen (512 × 512) corresponded to a compromise between image quality and acquisition time. All were acquired in stack mode and quantification performed on maximum intensity projections. The absence of potential biases induced by image acquisition at distinct resolution for fluorescence quantification has been verified as described in the next section.

For constructing *Figures 1*, *2* and *3*, *Figure 1—figure supplement 1* and *Figure 3—figure supplement 1*, as well as for selecting all cells where the fluorescence has been quantified, raw images were processed using ImageJ software version 1.50c4. The overall intensity of green and red signals was adjusted for each individual image up to the limit where the green and red auto-fluorescence of the cell wall, cytoplasm and nuclei of the xylem and mesophyll cells visually disappeared (the FBNSV being phloem-restricted, it does not invade xylem and mesophyll cells). Individual cells in phloem bundles were considered containing a segment when either green or red fluorescence (or both) could be visualized above background. Above-background fluorescence was never observed in non-phloem cells, consistent with the phloem restriction of this virus.

Infection by FBNSV distorts the anatomy of the phloem tissues and so distinguishing between phloem parenchyma cells, companion cells and sieve elements is difficult and sometimes impossible in our petiole cross-sections. For this reason, and because we were primarily interested by cells where the viral DNA replicates (ssDNA viruses replicate and accumulate in the nucleus of their host cells), we quantified the fluorescence only in those cells where a nucleus was clearly revealed by the DAPI staining, so in cells other than the sieve tubes. The DAPI staining allowed to precisely encircle the nucleus, and thus to define an ovoid corresponding area in each selected individual cell. Both green and red fluorescence were then quantified for each pixel within this area and the average pixel fluorescence intensity was estimated. This gave one average intensity value (in arbitrary fluorescence units) for green and one for red fluorescence in each selected individual cell within an infected petiole and all quantitative results are provided in *Supplementary file 2*: Table S4.

## Reliability of the quantification of fluorescence within individual cells of infected petioles

As indicated above, the setting of the confocal microscope was adjusted depending on the planned ulterior use of images. For example, to prepare *Figure 1*, the plane mode, higher resolution (>512×512) and higher magnification (63x objective) were generally preferred. In contrast, for fluorescence quantification, the stack mode and maximum intensity projection images appeared best appropriate in order to capture the whole fluorescence of the nuclei. For obvious practical reasons, the numerous images needed for quantification were acquired more rapidly at the feasible resolution of 512 × 512, and with a lower magnification (40x objective) in order to screen microscopy fields potentially allowing the scoring of more segment-containing cells (examples of such images are shown in *Figure 1B*, and *Figure 3A*). We wished to confirm that different resolution settings do not differentially affect green and red fluorescence. Another potential concern was the time spanning

between microscopy slide preparation and the end of the observation/quantification. Due to the experimental design, it was important to confirm that the stability of red and green probes was similar and that a possible differential degradation could not bias our localization and correlation/regression analysis.

We first quantified the fluorescence of R1-Green and R2-Red probes in petiole N°43 with fast acquired images (resolution 512 × 512). We then repeated the observation 2 weeks later on the same microscopy slides with a resolution 1024 × 1024. The cells quantified at the two dates were not necessarily the same as we could not be sure to retrieve all of them. The regression analysis from these two sets of images gave remarkably similar results, demonstrating that the resolution setting and the aging of the preparation could only have negligible effect (if at all) on our conclusions. Results from these controls are shown as *Figure 2—figure supplement 1*.

## Statistical analysis

Linear regression analyses were performed with the JMP 13.2.0 software. Each petiole was analyzed separately. We did not analyze the fluorescence quantification data set as a whole (pooling cells from distinct petioles) because the virus-associated fluorescence intensity is not directly comparable. The total viral load varies in between distinct petioles because of a 'natural' variance across infected plants and because some were harvested at different dpi. The auto-fluorescence of the infected tissues also varied across petioles because of the time of infection, the total viral load and presumably other unknown factors.

The proportion of cells containing both S and R segments, both S segment and M-Rep protein, or both R segment and M-Rep protein (*Figure 3C*, *Supplementary file 1*: Table S1) was analyzed with Generalized linear models (GLM) with 'treatment' (S segment – R segment/S segment – M-Rep/R segment – M-Rep) as a categorical explanatory factor, and a quasi-binomial error type. The GLM model was computed using R software 3.1.3. The name of each statistical test and its outcome is indicated in the text and figure legends.

## Data availability

All data are available in the manuscript and in Supplementary files.

Raw data of all quantified green and red fluorescence within individual cells of infected plants are provided as a separate Excel file, *Supplementary file 2*: Table S4.

To allow repeat/reproduce all correlation tests, the 508 raw/unprocessed images (.lsm format) used for preparing all figures and for fluorescence quantification in individual cells have been deposited in the public repository figshare. They can be accessed at the DOI: 10.6084/m9.figshare. 5981968.

## Acknowledgments

We are grateful to Sophie Leblaye for greenhouse work, plant production and virus inoculation. We also thank T Timchenko, B Gronenborn and J Vetten for kindly providing the antibody directed against the viral protein M-Rep. This work was supported by INRA dpt. SPE and by the grant 'Nano' N° ANR-14-CE02-0014 from the French national funding agency (ANR). MP, CU and SG acknowledge support from CIRAD. YM acknowledges support from IRD and CNRS.

## Additional information

### Funding

| Funder | Grant reference number | Author |
| --- | --- | --- |
| Institut National de la Recherche Agronomique | | Anne Sicard<br>Elodie Pirolles<br>Romain Gallet<br>Marie-Stéphanie Vernerey<br>Michel Yvon<br>Serafin Gutierrez<br>Stéphane Blanc |

| Centre National de la Re-cherche Scientifique | | Elodie Pirolles<br>Yannis Michalakis |
| Institut de Recherche pour le Développement | | Yannis Michalakis |
| Agence Nationale de la Re-cherche | ANR-14-CE02-0014 | Anne Sicard<br>Elodie Pirolles<br>Romain Gallet<br>Marie-Stéphanie Vernerey<br>Michel Yvon<br>Yannis Michalakis |

The funders had no role in study design, data collection and interpretation, or the decision to submit the work for publication.

## Author contributions

Anne Sicard, Conceptualization, Data curation, Investigation, Visualization, Methodology, Writing—review and editing; Elodie Pirolles, Formal analysis, Investigation, Methodology; Romain Gallet, Formal analysis, Methodology, Writing—review and editing; Marie-Stéphanie Vernerey, Data curation, Investigation, Methodology; Michel Yvon, Investigation, Methodology; Cica Urbino, Supervision, Investigation, Visualization, Methodology; Michel Peterschmitt, Investigation, Methodology, Writing—review and editing; Serafin Gutierrez, Conceptualization, Supervision, Investigation, Methodology, Writing—review and editing; Yannis Michalakis, Conceptualization, Formal analysis, Supervision, Funding acquisition, Validation, Investigation, Methodology, Writing—original draft, Project administration; Stéphane Blanc, Conceptualization, Data curation, Formal analysis, Supervision, Funding acquisition, Validation, Investigation, Methodology, Writing—original draft, Project administration, Writing—review and editing

## Author ORCIDs

Yannis Michalakis (iD) http://orcid.org/0000-0003-1929-0848
Stéphane Blanc (iD) http://orcid.org/0000-0002-3412-0989

## Decision letter and Author response

Decision letter https://doi.org/10.7554/eLife.43599.017
Author response https://doi.org/10.7554/eLife.43599.018

# Additional files

## Supplementary files

• Supplementary file 1. Supplementary Tables S1, S2 and S3.
DOI: https://doi.org/10.7554/eLife.43599.011

• Supplementary file 2. Supplementary Table S4.
DOI: https://doi.org/10.7554/eLife.43599.012

• Transparent reporting form
DOI: https://doi.org/10.7554/eLife.43599.013

## Data availability

All data are available in the manuscript and in Supplementary files. Raw data of all quantified green and red fluorescence within individual cells of infected plants are provided as a separate Excel supplementary file: Table S4. To allow repeat/reproduce all correlation tests, the 508 raw/unprocessed images (.lsm format) used for preparing all figures and for fluorescence quantification in individual cells have been deposited in the public repository figshare. They can be accessed at the DOI: 10.6084/m9.figshare.5981968

The following dataset was generated:

| Author(s) | Year | Dataset title | Dataset URL | Database and Identifier |
|---|---|---|---|---|
| Anne Sicard | 2018 | Sicard-2018-External-database-S1 | https://figshare.com/arti- | Figshare, 10.6084/m9. |

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
