## [Decision Letter]

Thank you for submitting your article "A multicellular way of life for a multipartite virus" for consideration by *eLife*. Your article has been reviewed by three peer reviewers, including Fernando García-Arenal as the guest Reviewing Editor and Reviewer #1, and the evaluation has been overseen by Detlef Weigel as the Senior Editor.

The reviewers have discussed the reviews with one another and the Reviewing Editor has drafted this decision to help you prepare a revised submission.

The paper has been reviewed now by three experts. All of them consider that the results are original, novel, relevant and important. All three reviewers, though, also consider that major claims, particularly proposing a new paradigm in (plant) virology based on a multicellular way of life, are not fully justified by the observations, and that the presentation and interpretation of the results should be toned down. This would not diminish the importance of the results, but it should help to highlight the observations that are new here. Below we summarise the major comments, as well as new experiments suggested that are strongly encouraged to sustain the present major claims.

1) The visualization of pairs of segments respectively labelled with green and red fluorochromes revealed that distinct segments do not necessarily co-occur but are most often found in different cells. Authors tested 7 out of the 28 possible pairs of segments. As shown in Table S1 in Supplementary file 1 the number of cells harbouring both segments is almost the same as the one describing the number of cells that more accumulates one of them. This cannot be an indication that a cell is mainly affected by one, and only one, segment. Also, the presented results, do not discard that in addition to the detected segment the rest of segments were at undetectable amounts in the same cell, a problem of which the authors are aware. Or it could happen that the mRNA was still active, which seems not to have been considered. In fact, the procedure for the FISH technique has a RNase treatment that eliminates RNAs present in the cell, thus, this possibility cannot be ruled out.

2) It is demonstrated that the function of a viral gene can act in a cell where the gene itself is not detected and these observations suggest that either the mRNA or the protein M-Rep itself can travel from the producing cells (those where segment R accumulates) to other cells of the host. Unfortunately, no evidence for such trafficking is given and thus, by the moment, is speculative. Authors must demonstrate that M-Rep behaves as a non-cellular autonomous protein or than its mRNA is able to traffic from cell to cell. Would the authors' result indicate that all the proteins encoded by FBNSV be non-cell-autonomous in their function or only some?

3) In the Discussion it should be considered the tissue tropism of FBNSV, and the particularities of plasmodesmata between the different phloem cell types, which may limit the generalisation of the presented observations to other viral systems.

These three points can be easily addressed in a revised Discussion section. A modification of the title would also be advisable.

4) The experiments demonstrate that segments of a unique type accumulate preferentially in individual cells, where they are found jointly with the replicase protein but not necessarily with the replicase gene. This observation, however, does not justify that a sufficiently high MOI is not needed and that the viral system is functional through complementation across cells. The experiments reported indicate a sort of specialization of different genes, such that one type is preferentially expressed in each cell. This might be a winner-takes-all strategy that does not necessarily entail isolation of the individual genes upon infection. Task allocation is an often used strategy (e.g. in social insects), and it would be great if it would have been discovered by multipartite viruses in order to optimize segment production. But, again, it would not solve the high MOI requirement for between-host propagation.

5) An experiment was proposed which would clarify these issues: Prepare two bacteria with only one type of segment (say R and S, as in most experiments performed). Inoculate a leaf at increasing distance with ONE copy of each segment. Demonstrate that the S segment does replicate. This experiment would answer two main concerns of the reviewers regarding the claims made: (i) Even if individual segments propagate independently and are affected by severe bottlenecks (such that they reach the plant in very low numbers), replication of any segment is possible; (ii) The amount of replicase protein is high enough so as to reach cells far apart and cause the replication of other "waiting" segments before either of them degrades.

6) Regarding the relationship between the experiments reported and the MOI required upon between-host transmission, two processes are apparently mixed here. The first one is between-plant transmission: as discussed above, this process requires an MOI so high for multipartite viruses with several segments that it has not been observed in nature: It has to be pointed out that the MOI as described e.g. in 5 has never been empirically quantified. Actually, what studies show is that the number of founding particles for the population infecting a given individual plant is low. But this is a number related to genealogy, not to transmission. (An analogous situation is population bottlenecks: the ancestry of a current population can be traced back to a few, even a single, individual, this fact not implying that the ancestral population was that small. Iranzo and Manrubia, 2012, concerns the size of the ancestral population, not the founding population.) The second one is the (theoretical and empirical) search of possible mechanisms that can alleviate that hypothetical high MOI. It has been put forward, with good empirical support, that viral particles might form groups or clusters upon propagation (and this might be relatively common).

7) Finally, there should be a discussion of the cost of producing very high amounts of mRNA and/or replicase protein that will be lost (as demonstrated by the experiments), if the interpretation of the authors proves to be correct.

---

## [Author Response]

The paper has been reviewed now by three experts. All of them consider that the results are original, novel, relevant and important. All three reviewers, though, also consider that major claims, particularly proposing a new paradigm in (plant) virology based on a multicellular way of life, are not fully justified by the observations, and that the presentation and interpretation of the results should be toned down. This would not diminish the importance of the results, but it should help to highlight the observations that are new here. Below we summarise the major comments, as well as new experiments suggested that are strongly encouraged to sustain the present major claims.

We agree that in various instances in the text of the original version of our manuscript, the writing was inappropriate and might have suggested that we claimed that the reported observation applied to all multipartite viruses and established a new paradigm in Virology; this was not our intention. To alleviate this problem, and to comply with the request from the reviewer to tone down this claim, we have modified those sentences to explicitly state that our discovery, at this point concerns the FBNSV infecting faba bean host plants.

More specifically:

In the Abstract the sentence “Our observation opens a new conceptual framework in virology where the infection operates at a level above the individual cell level, defining a viral multicellular way of life” is now replaced by the sentence “Our observation deviates from the classical conceptual framework in virology and opens an alternative possibility (at least for nanoviruses) where the infection can operate at a level above the individual cell level, defining a viral multicellular way of life”

At the end of the Introduction, the sentence “Using a highly multipartite virus with eight genome segments, we here suggest that the dominant conceptual framework in virology must be changed to comprehend these viral systems” is amended to “Using a highly multipartite virus with eight genome segments, we here propose that the conceptual framework in virology should be amended to account for such viral systems. Indeed, in our specific experimental model species, we demonstrate….way of life”.

In the Discussion, the sentence “Here we simply coin a new concept in virology, which is compatible with empirical observations and alleviates the question of the insurmountable cost in highly multipartite viral systems such as FBNSV…” is changed for “Here we introduce an additional concept in virology, which is compatible with empirical observations and which partially alleviates the insurmountable cost in highly multipartite viral systems such as FBNSV.”

1) The visualization of pairs of segments respectively labelled with green and red fluorochromes revealed that distinct segments do not necessarily co-occur but are most often found in different cells. Authors tested 7 out of the 28 possible pairs of segments. As shown in Table S1 in Supplementary file 1 the number of cells harbouring both segments is almost the same as the one describing the number of cells that more accumulates one of them. This cannot be an indication that a cell is mainly affected by one, and only one, segment. Also, the presented results, do not discard that in addition to the detected segment the rest of segments were at undetectable amounts in the same cell, a problem of which the authors are aware. Or it could happen that the mRNA was still active, which seems not to have been considered. In fact, the procedure for the FISH technique has a RNase treatment that eliminates RNAs present in the cell, thus, this possibility cannot be ruled out.

First, to be factual, it is not totally accurate to state “the number of cells harbouring both segments is almost the same as the one describing the number of cells that more accumulates one of them”. Figure 3C shows that, overall replicates, 42% of the cells contain the two segments of the pair R/S whereas 58% contain only one segment of the pair. From Table S1 one can calculate that these proportions are respectively 32% and 68% for the pair S/M, and are indeed more even for R/M with 46% and 54%.

We totally agree that this cannot indicate that the cell is infected by one and only one segment, and we do not think we have written it anywhere in the paper. This is an indication that the majority of the cells contain solely one of the segments of the tested pair, and that the cells that contain all 8 segments are most likely utterly rare. In fact an accurate estimate of the proportion of cells affected by all 8 segments is not feasible with our current data as it would require the investigation of numerous additional pairs of segments. However, it is clear from the three pairs quantified that the proportion of susceptible cells having the three segments S/R/M together is close to ~ 0.1, and thus that the proportion of cells containing all 8 segments should be much smaller. We conclude that the immense majority of the cells containing viral material do not contain the integral viral genome (all 8 segments). We have accordingly added a few words (in the Results) at the end of the last sentence of the paragraph “Remarkably, this applied even to the pairs from the three segments encoding for the three basic functions of plant viruses: replication (segment R encoding M-Rep), encapsidation (segment S encoding CP) and intra-host movement (segment M encoding MP), suggesting that cells containing all eight segments are extremely rare”.

That a segment could be present at undetectable levels is addressed and discussed in the paper, because all techniques obviously have their limitations. This is why we developed the fluorescence quantification approach, that informs on the interdependency of the distinct segments for their respective accumulation in individual cells and that is not affected by the detection limit; we assume this is the reason why this comment states “a problem of which the authors are aware”. So we are not sure whether this comment calls for a specific response on our side.

Because we can interpret the meaning of “mRNA was still active” in different ways, our response to the corresponding comment is plural. If it means that a mRNA could be present in a cell when the segment encoding it is not (and has never been present), then this is indeed a possibility related to mobile mRNAs [1-4] that is amply discussed in the manuscript. If it means that a DNA segment has been present in the cell but has disappeared while its mRNA is still there (and active), then it goes against our current understanding of how viral infections function. For all plant viruses investigated thus far, the genomic material is stored indefinitely within infected cells into mature viral particles (or in rarer cases as nucleo-protein complexes); the viral genome remains when the viral replication cycle is terminated and when viral mRNA (and translation) have long disappeared (for example see [5]). ssDNA viruses are believed to stably and indefinitely store their DNA as minichromosome (dsDNA) or as encapsidated ssDNA in the nucleus, and so are no exception to this rule (this is discussed with due references when explaining why some cells have the DNA segment accumulated but no detectable M-Rep protein)

Finally, we were aware that some mRNAs could be mobile [1-4] and so could be present in cells where their encoding DNA is not. This is why we used RNAse treatment when detecting and localizing DNA segments, because we did not want the possible presence of mobile mRNA to bias our interpretation on the localization of the DNA segments. To conclude, we do not rule out the possibility of the presence of mRNA in the cells that do not contain the cognate segment, this is even how we think the system could work (discussed in the paper when proposing the conceptual model)

2) It is demonstrated that the function of a viral gene can act in a cell where the gene itself is not detected and these observations suggest that either the mRNA or the protein M-Rep itself can travel from the producing cells (those where segment R accumulates) to other cells of the host. Unfortunately, no evidence for such trafficking is given and thus, by the moment, is speculative. Authors must demonstrate that M-Rep behaves as a non-cellular autonomous protein or than its mRNA is able to traffic from cell to cell. Would the authors' result indicate that all the proteins encoded by FBNSV be non-cell-autonomous in their function or only some?

Indeed, we do not know whether it is the mRNA or the M-Rep protein itself that travels to non-producing cells. The direct demonstration of the movement of the mRNA or the protein M-Rep is clearly part of our future projects. Unfortunately, to develop approaches other than the detection of DNA or proteins proposed here, we need to rely on tools that are not available in faba bean. We can e.g. imagine transient expression of a fusion M-Rep-GFP in identified cells and monitor its traffic to neighboring cells, or express the corresponding mRNA with a root- or companion cell-specific promotor in transgenic plants and detect it in distant green tissues. In both cases the corresponding tools are available solely for model plant species (i.e. *A. thaliana* or other model plants) that are non-host to FBNSV and where nanovirus gene expression products may thus not be fully functional. All relevant approaches for such direct demonstration will engender months/years of additional work and are in the end extremely “high risk” because currently implementable only in host species or tissues that are not naturally invaded by nanoviruses. We are clearly eager to do this research effort, but not reasonably for the present submitted short report (also see additional experiments in response to comment point 5 below).

Most importantly, we cannot fully agree with the statement that the movement of either the mRNA or the M-Rep protein is speculative at this point and that we have no evidence for it. Though indirect, we do provide two strong pieces of evidence: i) the presence of the protein M-Rep in numerous cells where the R segment is not detected and, perhaps the more convincing, ii) the demonstration that the protein M-Rep is not more associated with cells accumulating its encoding segment R than with cells which do not accumulate R but other segments. To stress that this is indirect evidence, we have accordingly amended the Result section by adding a few words in the last sentence of the section: “Although they represent indirect evidence, these observations together further support our conclusion that either the mRNA or the protein M-Rep itself can travel from the producing cells (those where segment R accumulates) to other cells of the host, as discussed below.”

It is not clear to us whether the last question, “Would the authors' result indicate that all the proteins encoded by FBNSV be non-cell-autonomous in their function or only some”, refers to our argumentation that a function may be present in cells where the genetic information is absent, or whether the question is more specific and asks whether all proteins (or their corresponding mRNA) need to be self-mobile or not. Whatever the question is, we do not know the answer; we can reply however that it is not mandatory that all proteins (or mRNAs) are non-cell-autonomous, though obviously the alleviation of the within-host putative cost of multipartitism through the non-cell-autonomous functioning is positively correlated to the number of non-cell-autonomous segment’s functions. Concerning the more specific question of the potential self-mobility, without a precise description of the mode of action of all viral genes, it is impossible to predict whether each of them should encode a self-mobile function. If the effector of a viral function needs to co-act or physically interact with the effector of another viral function in individual cells, then at least one of them should be mobile (self-mobile or transported by a carrier). In contrast, if a viral gene function is the triggering of an endogenous systemic plant response (e.g. hormone signal) that can propagate in plant tissues, then the self-mobility of this viral gene effector may not be required. At this point, we have no elements to soundly speculate on these numerous possibilities.

3) In the Discussion it should be considered the tissue tropism of FBNSV, and the particularities of plasmodesmata between the different phloem cell types, which may limit the generalisation of the presented observations to other viral systems.These three points can be easily addressed in a revised Discussion section. A modification of the title would also be advisable.

We are aware that the connections between companion cells (CC) and sieve elements (SE) are different than those between other tissues of the plant. The size exclusion limit (SEL) of the plasmodesmata (PD) is close to 60 KDa between CC and SE and potentially much smaller in other tissues [6,7]. All FBNSV protein monomers are small enough to freely traffic between CC and SE but this may not be true in other tissues. We thus understand the concern of the reviewers regarding the generalization of the presented observation, particularly to other multipartite viruses that are not phloem-limited and invade mesophyll and epidermis. Plants are capable of creating symplastic domains within which the cells are connected by widely opened plasmodesmata where large so-called non-cell-autonomous molecules and macromolecular complexes can freely traffic ^7^. These domains can appear and disappear during development and more specifically during organ formation and differentiation [7-9]. It is not impossible to imagine that some viruses could manipulate this plant inherent capacity and induce the appearance of symplastic domains where they could freely exchange viral products across distinct cells. Perhaps more directly relevant is the fact that all plant viruses produce movement proteins (MP) that could create symplastic domains. It is well and long established that viral MPs are non-cell-autonomous proteins that move away from the producing cells and enlarge the SEL of PD in mesophyll or epidermal cells [10]. This phenomenon has been frequently reviewed in the last decades and the movement profiles of MPs can even be quantified [11]. The point we want to make here is that, although we do not know how far our observations will be generalizable, there is no theoretical impediment to intercellular exchanges of viral material (proteins or mRNAs) in non-phloem tissues. At this stage, this possibility is pure speculation and this is why we initially chose not to discuss it. However, to comply with this comment we have inserted the following sentence in the discussion of the revised version: “Likewise, whether this non-cell-autonomous model can be extended to viral species invading non-phloem tissues where cell communication is more restricted is unknown. Non phloem-restricted viruses could induce the formation of symplastic domains, either by manipulating the endogenous capacity of the host plant to do so [7] or by opening plasmodesmata through the action of their non-cell-autonomous movement protein [10], but this possibility awaits further investigation.”

The last part of this reviewers’ comment is suggesting to also modify the title. While we agree to tone down our claims in several instances along the text (see our related modifications in response to the General comments above), we believe the title is adequately toned and we would like to maintain it as is. Indeed, it illustrates our observation and explicitly states that this observation is reported in one case, “for a multipartite virus”.

4) The experiments demonstrate that segments of a unique type accumulate preferentially in individual cells, where they are found jointly with the replicase protein but not necessarily with the replicase gene. This observation, however, does not justify that a sufficiently high MOI is not needed and that the viral system is functional through complementation across cells. The experiments reported indicate a sort of specialization of different genes, such that one type is preferentially expressed in each cell. This might be a winner-takes-all strategy that does not necessarily entail isolation of the individual genes upon infection. Task allocation is an often used strategy (e.g. in social insects), and it would be great if it would have been discovered by multipartite viruses in order to optimize segment production. But, again, it would not solve the high MOI requirement for between-host propagation.

The winner-takes-all strategy is proposed as an alternative view that we did not consider before, and we thank the reviewers for this comment. If we understand the comment properly the reviewer hypothesis is that the MOI could be high, with potentially several copies of each segment type entering within individual cells, but a ‘winner’ would replicate predominantly within each cell; the reviewer comment is not explicit on what the ‘winner’ would be and we see two possibilities that we discuss in turn: (i) the winner is the type of segment, e.g. R or S etc., and thus all molecules of this segment type initially present within a cell can replicate or (ii) the winner is a single molecule, of some segment type, that will engender all offspring within a cell through replication, other molecules of the same or different segment type initially present within this cell being left aside.

For the case (i) we ought to remind that all FBNSV segment types have a very conserved replication origin and that selectively replicating one segment type only in a cell, and another segment type only in another cell, is hardly conceivable (unless solely this type is present, as we propose). Moreover, our experiment with the genetic markers inserted in segment S or N pleads against this possibility: if the MOI was high, following the reviewers’ hypothesis, the observation that most cells accumulate only one of the alleles of each segment would not only imply the existence of a specific replication mechanism, but the specificity would need to operate at the allele and not the segment level; we fail to see any basis for such specificity and think that it is much more parsimonious to conclude, on the basis of our observations, that the MOI is low.

For the case (ii), assuming a high cellular MOI implies that many segments, even many copies of each segment type, are present and available in each cell but that a single one will efficiently replicate (the winner in the reviewers’ view). Again, all segments having the same replication origin we do not see how a single copy could be the only replication target, unless the M-Rep and replication complexes are in extremely limiting number, down to one. When looking at the distribution of M-Rep protein all over the nucleus (Figure 3), it is evident that many more than one copy is present. Nevertheless if we are to play an intellectual game, we could argue from our observation that the winner appears to be “designated” randomly: the segment type accumulating differs from cell to cell (Figure 1 and Figure 1—figure supplement 1) and the frequency of a segment in a tissue correlates with the number of cells where it is the winner (where it accumulates; Figure 2D). Would this be true, stochasticity in the initial stages of a plant infection would greatly determine the relative amount of the distinct segments accumulating later and so the genome formula. We have previously published that the genome formula is highly constrained to a reproducible pattern that is not dependent on the initial segment frequency, a fact that is not compatible with this speculation. This argument also applies to case (i) above.

In conclusion, whatever scenario we can think of for the winner-takes-all proposition, it is hard to reconcile it with what is observed on the biology of FBNSV both in this and in earlier published work.

The last sentence of this comment questions the transmission between hosts and is addressed in our response to the comment point 6 below.

5) An experiment was proposed which would clarify these issues: Prepare two bacteria with only one type of segment (say R and S, as in most experiments performed). Inoculate a leaf at increasing distance with ONE copy of each segment. Demonstrate that the S segment does replicate. This experiment would answer two main concerns of the reviewers regarding the claims made: (i) Even if individual segments propagate independently and are affected by severe bottlenecks (such that they reach the plant in very low numbers), replication of any segment is possible; (ii) The amount of replicase protein is high enough so as to reach cells far apart and cause the replication of other "waiting" segments before either of them degrades.

Despite our demonstration that segments can massively accumulate in cells where the segment R encoding the replication function is not detectable, that the intensity of replication of segments S and/or M does not correlate with the amount of segment R accumulating in the same cell, that the protein M-Rep accumulates in cells where its encoding segment is not detectable, and that the protein M-Rep does not colocalize more with its encoding segment than with another segment (S), we understand from this comment that the reviewers would appreciate an additional proof of the absence of R in virus replication competent cells.

We considered this comment/suggestion extremely carefully. We have accordingly performed an additional experiment (see Author response image 1) and, unfortunately, it confirms that the approach proposed by the reviewers is not feasible. There were several a priori drawbacks for the suggested experiment:

– It is totally out of the virus infection context and so interpretation could be questionable

– The products of segments other than S and R could be mandatory for the trafficking process

– The distance at which across-cell complementation is efficient could be too small for this experiment

– For unknown reasons, FBNSV clone is infectious solely when agro-infiltrated in the stem, not in leaves

– Wounding in the infiltrated zone can induce auto-fluorescence

– The agrobacterium plasmid with inserted viral sequences could be present in the observed cell

– We have no idea of the plasmid copy number and related artefactual fluorescence in observed cells

– We have no idea of the number of cells initially agro-inoculated and/or infected, and whether they would be too rare to be efficiently retrieved.

To collectively evaluate all these potential drawbacks, we decided to simply agro-inoculate faba bean stems as described for plant infection (with agro-bacteria carrying all 8 segments). This way, we are sure the viral infection is initiated in the inoculated region, which we sectioned, fixed, labeled and observed in search for fluorescent S (green) and/or R (red) signal. Our reasoning was that if no fluorescence could be detected in the nucleus of individual cells of these agro-infiltrated stem-tissues, it would be evidence that the approach proposed cannot work. Author response image 1 shows wound-related auto-fluorescence (Panel b), attesting that we are in the relevant inoculated area. Panel (c), (d) and (e) show typical microscopy fields including phloem bundles where no fluorescent cells could be observed, whether at 3, 5 or 10 days post-inoculation, and despite observing 49, 25 and 37 sections for each date, respectively. Panel (a) shows a similar experiment but in systemically infected tissues (just as in Figure 1 of the manuscript), controlling for the quality of our fluorescent probes. In this control case (in 15 sections), 41 fluorescent cells could be detected: 16 green only, 20 red only, and 5 green/red. We conclude that the cells where the infection initially starts are much too rare to be retrieved in panels (c)(d)(e), like looking for a needle in a haystack. Should we observe hundreds of sections from stems inoculated as suggested by the reviewers, we may eventually find a couple of fluorescent cells (or not), but would such rare occurrence be more convincing than all results shown in the paper in systemically infected tissues?

**Author response image 1. respfig1:** Attempt at detecting viral segments S and R in agro-infiltrated region of the stem. Trunks of faba bean stems agro-infiltrated with a mixture of agro-bacteria containing the eight FBNSV segments were sectioned and FISH-processed exactly as described in the Materials and methods section. Green-labeled S segments and Red-labeled R segments are visible solely in the petiole sections from systemically infected upper part of the plant (**a**) and not in the agro-infiltrated section of the stem (b-e). The wound (**w**) inflicted by the agro-infiltration process induces autofluorescence of the plant tissue that appears as a line of green and red fluorescence at the boundary of the wound (**b**). No fluorescent cells could be observed at three distinct dates, 3, 5 or 10 dpi, for which representative images are respectively shown in (**c**), (**d**) and (**e**). Horizontal bar is 50 micrometers.

6) Regarding the relationship between the experiments reported and the MOI required upon between-host transmission, two processes are apparently mixed here. The first one is between-plant transmission: as discussed above, this process requires an MOI so high for multipartite viruses with several segments that it has not been observed in nature: It has to be pointed out that the MOI as described e.g. in 5 has never been empirically quantified. Actually, what studies show is that the number of founding particles for the population infecting a given individual plant is low. But this is a number related to genealogy, not to transmission. (An analogous situation is population bottlenecks: the ancestry of a current population can be traced back to a few, even a single, individual, this fact not implying that the ancestral population was that small. Iranzo and Manrubia, 2012, concerns the size of the ancestral population, not the founding population.) The second one is the (theoretical and empirical) search of possible mechanisms that can alleviate that hypothetical high MOI. It has been put forward, with good empirical support, that viral particles might form groups or clusters upon propagation (and this might be relatively common).

We are not sure we correctly understand this comment, which discusses the MOI upon between host transmission, a question that we do not address in this manuscript. Perhaps the wording in the Discussion section was not clear enough on this point and we clarified with an explicit sentence in the corresponding lines: “Here we introduce an additional concept in virology, which is compatible with empirical observations and which partially alleviates the insurmountable cost in highly multipartite viral systems such as FBNSV: because concomitant infection of individual cells by all genomic segments is not necessary, the associated putative cost should be much smaller if not nil at the within host level. We earlier discussed the fact that the analogous cost upon between-host transmission and the mechanisms of its compensation remain to be understood [12].”

Despite this not being the object of the current submitted manuscript, concerning the specific comment that “*…*this is a number related to genealogy, not to transmission”, we wish to mention that one of our estimation methods in a recent paper discussing the plant-to-plant transmission of FBNSV by aphid vectors ^12^ relies entirely on transmission, and not at all on genealogy, and nevertheless indicates a relatively low founding population when a plant is inoculated by a single or by ten aphids. So it is possible that there is a small MOI in between hosts and, as discussed in [12], we do not know yet how the virus efficiently overcomes this problem.

Regarding the two last sentences of this comment, our results indicate that within host high MOI is not required for FBNSV because the segments do not need be together in every infected individual cell for the viral system to be functional. We demonstrate that the accumulation of one segment is independent of the accumulation of the others. Thus, observations in our viral system do not provide any support for (even go against) the hypothesis that groups/clusters of viral particles propagate ensuring that all segments always co-infect individual cells.

7) Finally, there should be a discussion of the cost of producing very high amounts of mRNA and/or replicase protein that will be lost (as demonstrated by the experiments), if the interpretation of the authors proves to be correct.

This comment is based on assumptions from the reviewers that are questionable:

First, our interpretation is that most susceptible cells where the mRNA/protein M-Rep will move can potentially contain a FBNSV segment and so does not necessarily induce a waste.

Second, producing viral material that will be lost is not an issue that is specifically relevant to our study, and it may even not be relevant in virology. Indeed, all viruses “waste” a huge proportion of whatever viral material is produced in the infected host. If we adopt the view expressed in this comment, wasting appears as the general lifestyle of viruses. For example, viruses build up populations of billions of particles within hosts, but only a few units are transmitted to next hosts in most cases [13]. Infected cells (whatever bacterial, animal or plant cells), in addition to numerous virus particles, also accumulate inclusion bodies that are often considered as storage sites for excess material, and this is left behind just as most of the virions produced which will not infect other cells.

Third, whether and how all these “wastes” translate as a cost for the virus is unclear because they are produced by the infected host. But this is a debate that is very far from the message of our paper and we would like to avoid entering it.

In summary, if agreed with the editors and reviewers, we would prefer not to discuss this point, because i) we are not sure our interpretation generates more waste than any other interpretation, ii) it is not a question addressed in the paper, iii) this question is not more acute with the here-reported peculiar FBNSV way of life than with any other described viral cycle.

List of cited references:

1 Winter, N. & Kragler, F. Conceptual and Methodological Considerations on mRNA And Proteins as Intercellular and Long-distance Signals. *Plant & cell physiology*, doi:10.1093/pcp/pcy140 (2018).

2 Reagan, B. C., Ganusova, E. E., Fernandez, J. C., McCray, T. N. & Burch-Smith, T. M. RNA on the move: The plasmodesmata perspective. *Plant Sci***275**, 1-10, doi:10.1016/j.plantsci.2018.07.001 (2018).

3 Liu, L. & Chen, X. Intercellular and systemic trafficking of RNAs in plants. *Nat Plants***4**, 869-878, doi:10.1038/s41477-018-0288-5 (2018).

4 Kehr, J. & Kragler, F. Long distance RNA movement. *The New phytologist***218**, 29-40, doi:10.1111/nph.15025 (2018).

5 Gutierrez, S. *et al.* The Multiplicity of Cellular Infection Changes Depending on the Route of Cell Infection in a Plant Virus. *J Virol***89**, 9665-9675, doi:10.1128/JVI.00537-15 (2015).

6 Knox, K. & Oparka, K. Illuminating the translocation stream. *Current opinion in plant biology***43**, 113-118, doi:10.1016/j.pbi.2018.03.009 (2018).

7 Faulkner, C. Plasmodesmata and the symplast. *Curr Biol***28**, R1374-R1378, doi:10.1016/j.cub.2018.11.004 (2018).

8 Otero, S., Helariutta, Y. & Benitez-Alfonso, Y. Symplastic communication in organ formation and tissue patterning. *Current opinion in plant biology***29**, 21-28, doi:10.1016/j.pbi.2015.10.007 (2016).

9 Tilsner, J., Nicolas, W., Rosado, A. & Bayer, E. M. Staying Tight: Plasmodesmal Membrane Contact Sites and the Control of Cell-to-Cell Connectivity in Plants. *Annual review of plant biology***67**, 337-364, doi:10.1146/annurev-arplant-043015-111840 (2016).

10 Lucas, W. J. Plant viral movement proteins: agents for cell-to-cell trafficking of viral genomes. *Virology***344**, 169-184, doi:10.1016/j.virol.2005.09.026 (2006).

11 Trutnyeva, K., Ruggenthaler, P. & Waigmann, E. Movement profiles: a tool for quantitative analysis of cell-to-cell movement of plant viral movement proteins. *Methods in molecular biology (Clifton, N.J***451**, 317-329, doi:10.1007/978-1-59745-102-4_22 (2008).

12 Gallet, R. *et al.* Small Bottleneck Size in a Highly Multipartite Virus during a Complete Infection Cycle. *J Virol***92**, doi:10.1128/JVI.00139-18 (2018).

13 Gutierrez, S., Michalakis, Y. & Blanc, S. Virus population bottlenecks during within-host progression and host-to-host transmission. *Current opinion in virology***2**, 546-555 (2012).